# Facilitators and barriers to uptake of digital adherence technologies in improving TB care in Ethiopia: A qualitative study

Zewdneh Shewamene[1]*, Mahilet Belachew[2], Amanuel Shiferaw[2], Liza De Groot[3], Mamush Sahlie[2], Demekech Gadissa[2], Tofik Abdurhman[2], Ahmed Bedru[2], Taye Leta[4], Tanyaradzwa Dube[5], Natasha Deyanova[6], Degu Jerene[3], Katherine Fielding[1], Amare W. Tadesse[1]

1 London School of Hygiene & Tropical Medicine, London, United Kingdom, 2 KNCV Tuberculosis Foundation, Addis Ababa, Ethiopia, 3 KNCV Tuberculosis Foundation, The Hague, Netherlands, 4 National Tuberculosis Control Program, Ethiopian Ministry of Health, Addis Ababa, Ethiopia, 5 The Aurum Institute, Johannesburg, South Africa, 6 Organization for Appropriate Technologies in Health, Ukraine

* Zeedshow@gmail.com

**Data Availability Statement:** All relevant data are within the manuscript and its Supporting Information files.

## Abstract

The role of digital adherence technologies (DATs) in improving tuberculosis (TB) treatment adherence is an emerging area of policy discussion. Given that the directly observed therapy (DOT) has several shortcomings, alternative approaches such as DATs are vital to enhancing current practices by rendering person-centered models to support the completion of TB treatments. However, there is a lack of evidence that informs policy and program on facilitators and barriers to the uptake of DATs in the context of country-specific real-world situations. The purpose of this study was to explore the facilitators and barriers to the uptake of DATs by drawing from the accounts of people with TB (PWTB), healthcare workers (HCWs) and other key policy stakeholders in Ethiopia. A qualitative study was conducted to capture the perspectives of participants to help understand the contextual factors that are important in the uptake of DATs. The overall response from participants highlighted that uptake of DATs was high despite some critical implementation barriers. DATs were useful in reducing the burden of treatment management on both PWTB and HCWs, improving adherence and flexibility, and enhancing the patient-provider relationship. The relative simplicity of using DATs, positive feedback from important others, and current policy opportunities were seen as additional facilitators for the uptake of DATs in the Ethiopian context. Key barriers including network issues (mobile phone signals), lack of inclusivity and fear of stigma (as perceived by HCWs) were identified as key barriers that could limit the implementation of DATs. The findings of this qualitative study have provided a rich set of perspectives relevant to policymakers, providers and implementers in identifying the facilitators and barriers to the uptake of DATs in Ethiopia. The overall finding suggests that DATs are highly acceptable among the diverse categories of participants in the presence of critical barriers that limit uptake of DATs including poor infrastructure. However, key policy stakeholders believe that there are several opportunities and initiatives for feasible implementation, adaptation and scale-up of DATs in the current Ethiopian context.

**Funding:** This study was supported by Unitaid (Grant # 2019–33-ASCENT to the KNCV Tuberculosis Foundation) through the Adherence Support Coalition to End TB (ASCENT) project https://www.digitaladherence.org/. The funder of the study had no role in the collection, analysis, interpretation of data, and decision to submit the results for publication.

**Competing interests:** The authors have declared that no competing interests exist.

## Author summary

Improving tuberculosis treatment adherence is key to decreasing the risk of poor treatment outcomes. The traditional facility-based directly observed therapy (DOT) has limitations including frequent absence from work due to the need to visit health facilities daily, lack of transportation and the associated cost, and loss of autonomy. This approach also assumes that all people with tuberculosis require the same level of monitoring and support, rather than providing person-centered care based on specific adherence profiles. Digital adherence technologies (DATs), including the smart pillbox and medication labels, have emerged as an alternative to DOT to improve treatment adherence. We have implemented two DATs (smart pillboxes and medication labels) in Ethiopia and qualitatively examined the uptake of these technologies among people with TB, healthcare workers, and key stakeholders. Respondents have identified that DATs are relevant to improving adherence by reducing burdens related to frequent health facility visits, enhancing person-centered care, and increasing flexibility.

## Introduction

Despite tuberculosis (TB) being preventable and curable, it remains the first leading infectious killer, except for the year 2022 when the coronavirus (COVID-19) pandemic became the primary cause of death [1]. While the facility-based directly observed therapy (DOT) approach has achieved significant improvements in TB treatment outcomes since its introduction, global success rates remain substantially below the target for the 2025 End TB strategy [2,3].

DOT involves a healthcare provider or trained individual observing and recording a patient taking their medication to ensure adherence, and patients usually need to visit a healthcare facility for this process daily. The implementation of facility-based DOT places a significant burden on people with TB (PWTB) including stigma, absence from work due to the need to visit health facilities, lack of transportation and the associated cost, and loss of autonomy [2,4,5]. Effective implementation of DOT also requires a huge investment, which accounts for up to 75% of the provider costs of TB treatment (4) and adds more burden to the busy healthcare facilities [2,5]. Another shortcoming of the DOT approach is the assumption that all PWTB require the same level of monitoring and support, rather than providing differentiated care based on specific adherence profiles [6,7].

TB remains a significant public health challenge in Ethiopia. The World Health Organization's Global Tuberculosis Report 2023 highlights Ethiopia as one of the high TB burden countries, with an estimated incidence rate of 151 per 100,000 population [1]. The country has adopted facility-based DOT for drug-sensitive TB since 1997 as its core treatment approach and endorsed several global strategies such as global Stop TB and End TB strategies [8,9]. Despite a high treatment success rate reported over several consecutive years, poorer treatment outcomes and inconsistent implementation of the DOT posed a major challenge in TB elimination [10].

Given the limitations of traditional facility-based DOT strategy, alternative approaches such as digital adherence technologies (DATs) are emerging to enhance current practices by rendering person-centered models to support completion of TB treatments [11–13]. However, it is not clear how DATs could work within local routine-care contexts to help policymakers and providers effectively and sustainably implement such technologies on a larger scale. As such, the purpose of this study is to explore the facilitators and barriers to the uptake of DATs

among PWTB, health providers and key stakeholders in Ethiopia. A qualitative method was chosen to best capture and describe the highly contextual and behavioral characteristics that are important in the uptake of DATs.

## Methods

### Study setting

This study was nested within the "Adherence Support Coalition to End TB (ASCENT)" cluster-randomised trial in Ethiopia that aimed to evaluate the effectiveness of DATs with differentiated care to improve treatment outcomes for drug-sensitive TB. The trial was conducted in 78 health facilities in Addis Ababa and Oromia regions of Ethiopia [14]. Using the Ethiopian Government's definition of urban and rural, five of the 78 health facilities were located in rural areas, all of which were in Oromia.

Two DATs were being evaluated; smart pillboxes and medication labels with an associated adherence platform. Smart pillboxes had a daily audio-visual reminder for the PWTB to take treatment and were equipped with sensors and connectivity features that recorded when the pillbox was opened on the platform. The medication label, affixed to the fixed-dose blister-packaged TB medication, displayed a unique code that was valid for one week. Patients sent a text with these codes every day to confirm their daily dose was taken, with the platform logging data and sending reminders if doses were missed. The adherence platform was used by health providers in health facilities to monitor patients' adherence (using proxies of pillbox opening or SMS sent) in real time through a web-based dashboard. This platform enabled providers to track treatment progress, send reminders, and follow up with patients who miss doses, aiming to improve adherence and provide timely support [14].

### Study design and participants

Participants for the qualitative study were enrolled from purposively selected health facilities implementing the smart pillbox or medication label interventions from both regions (S1 Appendix). In-depth qualitative interviews were conducted with four categories of participants: 1) people with drug-sensitive TB (12 facilities), 2) people with multi-drug resistant TB (six facilities), 3) healthcare workers (20 facilities), and 4) other key stakeholders (policymakers, community members, and implementing partners). People with multi-drug resistant TB (MDR-TB) were not part of the randomised trial, though they did receive the smart pillboxes, and followed up for treatment outcomes. They were included in the interviews to better understand their perspectives in terms of the acceptability of DATs and their usefulness in improving adherence in MDR-TB settings. Interviews were conducted between March 2022 and February 2023.

Specific inclusion criteria were used to select PWTB and HCWs including adherence profile, age, sex, geography, and length of DAT use. Key stakeholders were purposively selected from the national TB program, implementation partners and the community considering their engagement and experience in this study and the national TB care policy in Ethiopia. Separate semi-structured interview guides were developed for each group of participants by the research team (S2 Appendix). The interviews with PWTB and healthcare providers were conducted in private rooms at health facilities, while stakeholders were interviewed at their offices. Saturation was declared when no new information was obtained.

### Interview process

In-depth interviews using semi-structured topic guides were conducted by two research assistants (male and female) who have BSc degrees in public health. The research assistants received

one day of training on qualitative interview techniques, the consenting process, and the contextual factors that interviewers need to consider to ensure privacy and cultural appropriateness while greeting and talking to persons with TB. At the end of the training, the research assistants conducted practice interviews among themselves to practice and refine their skills. Feedback was provided by the trainers, ensuring that any misunderstandings or errors were promptly addressed. This was followed by several discussion sessions and debriefings with a senior qualitative researcher throughout the data collection process to fine-tune the topic guides and ensure data quality. PWTB were interviewed at health facilities during their refill visit based on their treatment follow-up appointments. PWTB were pre-informed by TB focal health professionals about the interview and invited to voluntarily participate in the study.

## Data analysis

All interviews were digitally audio-recorded and were conducted in local languages (Amharic and Afaan Oromo), translated and transcribed verbatim to English by experienced researchers who were recruited for this purpose (PhD students in public health). The primary author evaluated all original audio recordings against transcripts to ensure the accuracy and completeness of translations. Analysis initially included familiarisation and discussion of the interviews. The researchers coded interview transcripts inductively in NVivo and agreed on the preliminary coding framework. To ensure inter-rater reliability, we used double coding of the first six interviews (ZS, LDG), followed by discussion and consensus meetings to resolve any discrepancies. The first author coded all the remaining transcripts, and a final coding framework was developed. A codebook was generated and shared with the entire research team and key themes and interpretations were identified.

The interview guides and the qualitative analysis were informed by key constructs of the Unified Theory of Acceptance and Use of Technology (UTAUT) framework to examine the acceptability and uptake of DATs. The UTAUT framework has been used in previous studies and posits that acceptance of a new technology can be determined by the effects of performance expectancy (usefulness), effort expectancy (easy to use), social influence and facilitating conditions [15–17].

Sequential numbers and the category of respondents were used to save transcripts and audio recordings. To protect the identity of respondents, the names of participants, institutions, and individuals mentioned during the interviews have not been reported. Research interviews and transcripts were digitally stored securely on a password-protected network drive to protect the confidentiality and privacy of participants. Research interviews and transcripts will be deleted after ten years according to the data retention policies specified in the research protocol.

Direct quotes extracted from interviews are identified using the category of the respondent and numbers to avoid accidental breach of anonymity (e.g., PWTB #1, HCW #2, stakeholder #3, etc.). All personal identifiers were removed from all records to ensure confidentiality.

## Ethical consideration

Ethical clearance for the ASCENT study which included the trial and qualitative sub-studies was obtained from the London School of Hygiene & Tropical Medicine Ethics Committee, United Kingdom (19120–1) and the WHO Ethical Review Committee, Switzerland (0003297). In addition, ethical approval was granted from the two regional health bureaus in Ethiopia where the study was undertaken: Addis Ababa City Administration Health Bureau (A/A/H/B/ 1/978/227) and Oromia Regional Health Bureau (BEFO/HBTFH/1-16/10322). All participants provided both written informed consents to participate in the study and for the interviews to

be recorded. Those who could not read and write provided a thumbprint as a form of signature.

# Results

## Characteristics of participants

In total, 57 participants were interviewed across four categories: 25 people with drug-sensitive TB, eight people with MDR-TB, 20 healthcare workers, and 4 national stakeholders in TB care (S1 Appendix). Of the 25 participants with drug-sensitive TB, 13 were using medication labels, and 12 were using the smart pillbox. Nearly equal number of female and male participants (13/12) participated, with an average age of 34 years. People with MDR-TB (n = 8; 4 female and 4 male) were recruited from six health facilities in Addis Ababa and all were receiving the smart pillbox intervention. Healthcare workers (n = 20) were selected from health facilities who were implementing DATs (10 participants each from the label and smart pillbox arms). The majority of healthcare workers were female nurses (n = 15). The four national stakeholders were purposively selected from the Ministry of Health (MOH), implementing partners and TB survivors, considering their level of engagement in the Ethiopian TB program.

## Facilitators and barriers to uptake of DATs

Overall, responses from all groups of participants ascertained that DATs were highly acceptable despite the existence of critical implementation barriers.

Informed by the UTAUT framework, the perspectives across all types of participants identified four broader themes that capture the facilitators for the uptake of DATs in the current Ethiopian context (Fig 1). First, DATs were considered very useful for both PWTB and HCWs (**Performance Expectancy**). Second, both PWTB and HCWs indicated that, practically DATs were easier to use than their pre-expectation before the intervention was started (**Effort Expectancy**). Third, PWTB revealed that the responses they received from important others about DATs were positive and encouraged them to use it (**Social Influence**). Fourth, there are promising policy opportunities and initiatives to increase the uptake of DATs in Ethiopia including strong TB care structures and digitization initiatives by MOH, and expansion of the telecom services (**Facilitating Conditions**). On the other hand, barriers to DAT uptake were constructed into three themes containing the critical factors limiting the implementation of DATs. These key uptake challenges include network issues, lack of inclusivity, and perceived stigma (Fig 1). The overall findings highlighted that there were no differences by age and sex in the context of experiences and perspectives of using DATs among the study respondents.

## Facilitators of DATs uptake

Four themes were identified on the facilitators of DATs uptake.

### Theme 1: Usefulness of DATs (facilitator)

Participants noted that DATs were useful in reducing the burden of daily health facility visits (saving time and reducing cost), improving treatment adherence and flexibility, and enhancing patient-provider relationships. From the healthcare perspective, DATs were useful in reducing their workload, work-related TB exposure, enabling them to provide differentiated responses (person-centred care), and increasing their motivation in providing care to PWTB.

**Reduce frequency of health facility visits.** One of the most notable advantages of using DAT as described by PWTB was its ability to reduce the frequency of health facility visits. While routine TB care requires PWTB to collect their medication from the health facility daily,

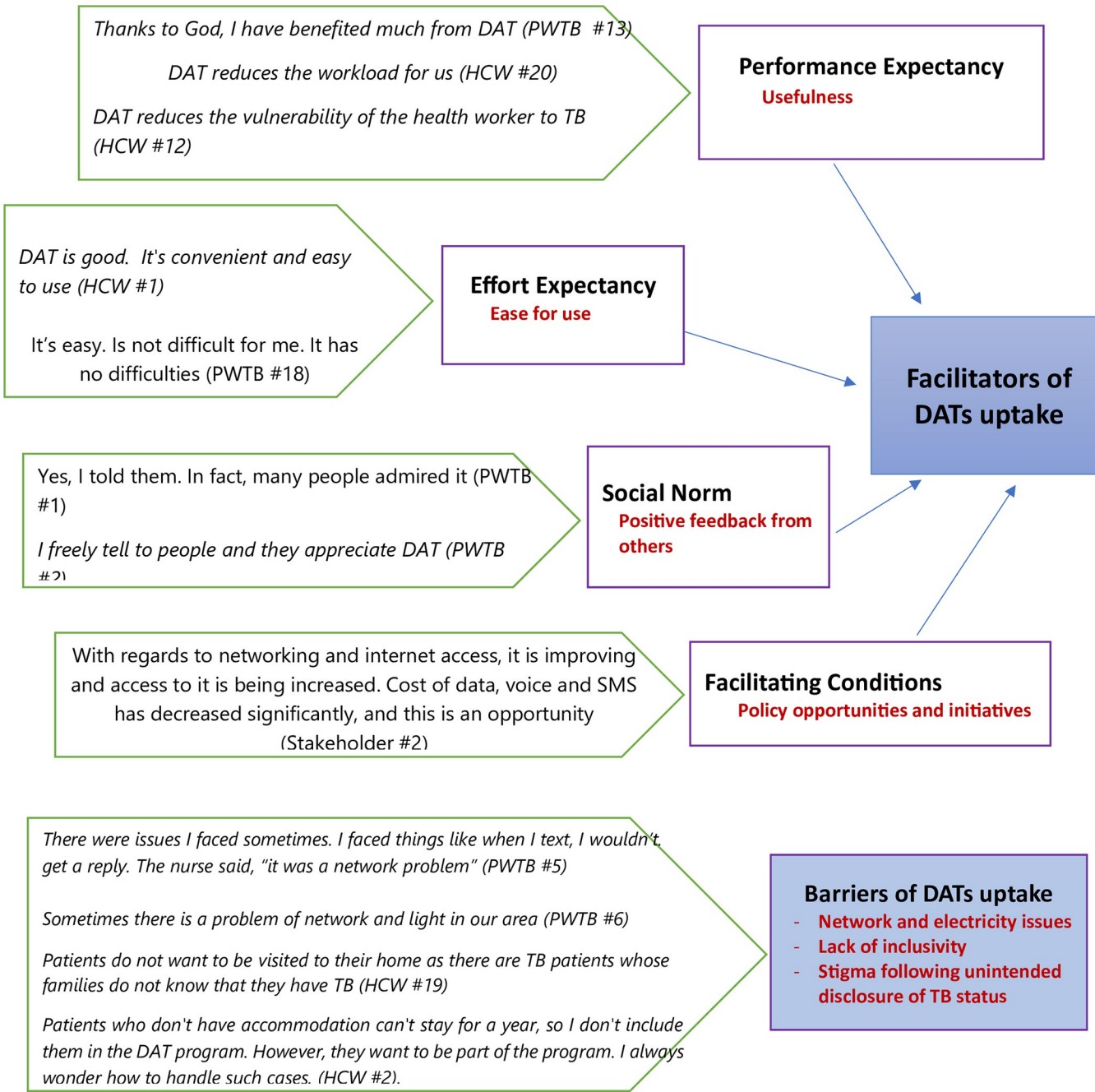

**Fig 1. Facilitators and barriers of DATs uptake in Ethiopia (adapted from Thomas et.al 2021).**

PWTB who enrolled into the DATs can visit fortnightly or monthly. The platform assisted health professionals in monitoring adherence and providing differentiated care based on the adherence profiles of each PWTB.

> *Now, I am supposed to go to health center twice in a month. Previously, the health center workers used to force patients to come and take the medicine every day especially at the beginning of the medication [DOT phase]. The smart pillbox has eased all these challenges. (PWTB # 11, Smart Pillbox Arm)*

*It [Medication Label] helped me very much. Travelling from there [home] to here [health facility] is tiresome. If I was collecting my medication by travelling to the health facility every day, I could have stopped taking it. (PWTB # 13, Medication Label Arm)*

*It [DAT] reduces frequent travel to the facility. Previously patients have stopped coming due to frequent travel to the facility. (HCW #20).*

PWTB considered the DATs as vital tools to save their time. Otherwise, they waste several hours each day taking their medications at health facilities. In settings like Ethiopia, where heavy traffic jams in bigger cities and the unavailability of public transport in rural areas are bitter realities, saving daily travel time can significantly improve participants' productivity. For example, respondents summarized their experience with the DATs as:

*If I must come daily [to the health facility], it's impossible to work. For sure taking the medicine will be interrupted. If you tell me to come and take it [medication] daily, it won't go along with my work. It can't work. I missed work to come here today [on the interview day]. I requested a half day leave to come here. (PWTB #4, Medication Label Arm)*

*I have saved my time. For example, I must have to come to health center daily if there was no this smart pillbox. It means you are saving one day, which is not simple. (PWTB # 11, Smart Pillbox Arm)*

Removing the transportation cost of daily travel during DOT was also described as one of the many advantages of using DATs. Without DATs, participants' previous experience was negative due to having financial constraints for transportation. Some PWTB shared their experiences of walking a long distance daily to collect their medication because they were unable to pay for a taxi.

*Previously I used to walk if I didn't have money for taxi, but now you take your medication with you. You are told to take this amount for certain days at a certain time and you send a code with the new technology. if you didn't, it's like waking you up and making you take the medication. (PWTB #3, Medication Label Arm)*

*The benefit of DAT is specially in these times of high economic inflation where patients cannot travel to the facility regularly to take the treatment. (HCW #13)*

**Improved treatment adherence.**   Overall, PWTBs and HCWs have positive experiences in using DATs in terms of helping them take their medication timely due to the reminders function and differentiated model of care they received from health professionals when they missed *doses* (text, phone call and home visits).

*... it reminds me not to forget the medicine. When they send me a reminder, I remember to take the medicine. When I forget taking the medicine, I receive a message which says "please take your medicine, you didn't take the medicine". When I forgotten taking the medicine, the reminder helps me take it. (PWTB #12, Medication Label Arm)*

*They [PWTB] sometimes get fed up and drop out of the treatment. This technology [DAT] has increased the success of tuberculosis treatment. We have good results and good successes. (HCW #10)*

*This [DAT] helps the patient to take their medications at the right time. It is something important. I'm not saying DOT wasn't a correct model, but DAT has become much better. (HCW #16)*

Participants also shared their accounts of how their TB treatment adherence would have been without the help of DATs.

*If I had been collecting my medication by travelling every day to the health facility, I could have stopped taking it. (PWTB #13, Medication Label Arm)*

**Better treatment flexibility.** The DAT improved the freedom of PWTB in terms of "when and where" they can take their medication compared to the routine DOT, as described by participants.

*Now you can go wherever you want because the medicine is at your hand. You can take the medicine wherever you are. Therefore, it is very important, and it helped me very well. (PWTB # 12, Medication Label Arm)*

*It reduces burden of travelling to health facility every morning and it created an opportunity to use it in any setting or in my home. (PWTB #11, Medication Label Arm)*

*DATs make it possible to follow a patient from anywhere, including at night (HCW #8)*

**Improved patient-provider relationship through person-centered care.** Both PWTB and healthcare workers indicated that the DATs and Differentiated model of care were useful mechanisms to improve the relationship and the provision of person-centered TB care. In return, this positive relationship improved treatment adherence and satisfaction PWTB.

*I feel I am being followed up [by health professionals] and this follow up is very important for someone who has not taken the medication. It is very important reminder to take the medication at the right time. There are times I delayed taking my medication, but when I see the text, it reminds me, and I take it. (PWTB #4, Medication Label Arm)*

*It [DAT] helped in providing good patient care and increasing the relationship between healthcare professionals and the patient (HCW #10).*

The DAT helped PWTB receive more person-centered care from health workers. Most participants' views indicated that they were able to build a stronger relationship with HCWs and this has created the feeling of being supported which encouraged PWTB to take their medication.

*The DAT keeps me in touch with the health professionals. I feel she [the nurse] cares for me. She could have leave without calling but she calls me because she cares. For me what made the program [DAT and differentiated model of care] easier is the message service [part of the differentiated model of care]. It has brought the easiest way for me to take my treatment and medications. (PWTB #2, Medication Label Arm)*

On the other hand, HCWs said the DAT gave them the opportunity to closely monitor participants' adherence profile which enabled to provide a differentiated model of care. Based on

the information from the DAT, they were able to counsel PWTB through phone calls, SMS and in-person through the use of health extension workers.

*In my case, they're [Patients] very happy while I called them. They consider my phone call as big deal and increased more our intimacy (HCW #11).*

*This [DAT] improves our relationship. For example, many patients call me by the address number of the technology. I tell them to call me by my own phone number. We are almost family. When they see me in the street, they greet me and I know the patients who use this technology (HCW #12).*

**Reduced workload.**   HCWs perceived that implementing DATs reduces their workload. These technologies facilitate easier adherence monitoring, which enables HCWs to spend more quality time with patients who need extra support and decreases the overall demand on their time due to fewer required clinic visits by patients.

*For us it decreases the workload by providing the treatment once biweekly for patients and having a technology assisted follow up, rather than giving the service on a daily basis. (HCW #11).*

**Reduced work-related TB exposure.**   By decreasing the number of PWTB who need to visit the health facility for directly observed therapy, DATs also reduce the chances of HCWs being exposed to TB. This reduction in patient visits helps minimize the risk of infection for healthcare workers compared to the usual directly observed therapy. For instance, a respondent said:

*It's [DAT] major importance is it reduces risk of TB transmission. In the first two weeks in which there is high chance of transmission, you do not contact with patients daily unlike in DOT where you contact with them every day. In DOT you contact and observe patients daily which is risky for the health worker. In this regard, DAT is very important to reduce TB exposure (HCW #19).*

## Theme 2: Easy to use (facilitator)

The simplicity of using DATs was reflected repeatedly in responses from PWTB and HCWs. PWTB reported that taking their medication through the use of DATs did not pose major challenges.

*It's easy. Is not difficult for me, charging it also simple, it has no difficulties. (PWTB #18, Smart Pillbox Arm)*

*First, I it is very simple and helpful for me. (PWTB #18, Smart Pillbox Arm)*

Similarly, HCWs revealed that practical implementation of DATs was easier than their expectations during the initial training they received.

*DAT is good. Health professionals monitor patients from home while at the health center. Healthcare professionals can easily call patients and invite them or do follow-ups. (HCW #8).*

*I thought that it [DAT] was hard to implement. I even said an impossible thing is coming during the training for the first time. But gradually it becomes easy for me. I was considering it to be challenging and complicated for patients but now, it becomes an important thing (HCW #9).*

The initial orientation delivered by health professionals played a vital role in making use of DATs simple to PWTB. Overall, PWTB were satisfied with the initial orientation provided by HCWs about how and why to use the DAT.

*When I first arrived here [health facility], I met the health professional working at the TB clinic. she immediately explained to me the general description of the box that was completely new and that it was for testing. The explanation is enough for people like me [educated patients]. The information provided by the nurse was very helpful to me (PWTB #1; Smart Pillbox Arm).*

*It [the information provided during the orientation session] was enough. What she [the Nurse] told us was about texting, she adjusted my phone and I started texting. It was not very difficult. When you text, you immediately receives texts within seconds. so it was not very difficult to send or receive. it's very clear (PWTB #5, Medication Label Arm)*

While there were PWTB who did not fully understand the procedure during the initiation phase, they reported that healthcare providers were able to assist by providing continuous information to clarify issues.

*It is easy to use but I was worried about the battery because I was not told about it, but I finished my treatment before the smart pillbox run out of charge. If they told me the color of the light will be changed when it runs out of charge, I will be cautious (PWTB #6, Smart Pillbox Arm).*

*Indeed, she has told me many things. But I don't have clue how to send the message by the time [when the orientation was delivered]. I asked her [the nurse] how to send the message and she oriented me [the next day], then I started using [the label] the next day (PWTB #2, Medication Label Arm)*

PWTB using the smart pillbox also identified that DAT made it easier to properly keep their medicine at home and workplace.

*It [the smart pillbox] helps me put the medicine properly. It helps for the proper handling of the medicine. if you are using the medicine without the smart pillbox, you worry about where to put and the temperature. It has a potential to preserve the medicine. (PWTB #11, Smart Pillbox Arm)*

*It [the smart pillbox] protect the medicine from becoming dirty because you put it inside the box. (PWTB #24, Smart Pillbox Arm)*

### Theme 3: Positive feedback from important others (facilitator)

PWTB reflected that responses they received from important significant others (families, friends, colleagues and neighbors) were positive and encouraging for them to use DATs. While there were some reservations, most participants indicated that they were not afraid of telling other people about DATs and how they are useful in their TB treatment.

*Honestly, I don't have that complicated feeling to tell people [about DATs]. I freely tell to whosoever. People traditionally used to follow their treatment by going to the health center every day, but this medication label is quite different from that and I found it relevant. People appreciate the service for the reason that I don't have to go to health center every day (PWTB #2, Medication Label Arm)*

### Theme 4: Policy opportunities and initiatives (facilitator)

Stakeholders identified that there are several initiatives and opportunities which can increase the uptake of DATs in the Ethiopian context. For instance, the recent expansion of the telecom infrastructure in Ethiopia following the new agreement between an international provider and the Ethiopian government is anticipated to increase access to mobile phone networks. In this study there is no cost to the PWTB for the SMS sent in the label arm.

*Communication institutions like Safaricom are entering to our country and there will be cost minimization. So, through DAT can be scaled-up. The patient may be charged maximum 10 cents to send a message per day. (STK #4).*

*There is an expanding telecommunication through Safaricom. The CEO of Ethio-telecom is going to be an ambassador of TB program in the near future and this will be a good opportunity. Safaricom may reduce the cost of messaging. (STK #1).*

Stakeholders also identified that existing TB care structures and coordination platforms within the ministry of health (MOH) is vital to expand DAT implementation.

*There is a TB technical working group at national and regional level. If there is a focal person for DAT [within the technical groups] it will be more helpful [to integrate into national programs] (STK #1).*

The new national healthcare digitization initiatives was also seen as major opportunities that enhance uptake of DAT implementation in Ethiopia. A participant mentioned that a taskforce was established at MOH to lead the digitisation initiatives which will be important to support the feasibility and scale up of DAT implementation.

*The ministry of health also supports the digital technology implementation. To lead this digitisation, a task force in the ministry is established to be led by the state minister. So, bringing all the fragmented activities in the same pool is one opportunity for scaling up of technologies like DATs (STK #4).*

### Barriers to uptake of DATs

Responses across all groups of participants indicated that lack of adequate infrastructure including poor telecom signals and electricity supply are the major challenges for DATs implementation. Providers and stakeholders were also concerned that DATs may not be used by all PWTB. For example, PWTB with low literacy, those who did not have phones, and people who are homeless and prisoners may not have the opportunity to use DATs. Stigma was also indicated as a potential barrier to the implementation of DATs. HCWs have reflected their concern (perceived) that use of DATs may lead to unintended disclosure of TB status (e.g, during use of DATs at workplace or home visits by health workers) which in turn can result in the fear of stigma among PWTB. However, responses from PWTB did not support this concern (either perceived or experienced stigma) related to use of DATs.

### Theme 1: Network and electricity issues (barrier)

As both the smart pillbox and medication label adherence technologies are built to be linked with phone devices, poor telecommunication network and frequent power outages in Ethiopia posed a wide range of challenges in the effective utilisation of DATs. As a consequence of poor network or power outages, patients identified several types of challenges they were experiencing in the use of DATs. For example, patients experienced a delay in receiving text reminders or alarms because of network or power issues and their time to take the medication may become overdue.

*Sometimes it is quiet, does not give alert even if time is up, I always watch my phone for time, so I take my medicine timely even if the smart pillbox does not give any alert. [PWTB #6, Smart Pillbox Arm)*

*Sometimes there is a problem of network and light in our area. I can take the medicine without the alarm by checking the time from my watch. When I check its alarm [alarm of the smart poll box], the time is up [but not alert], I come to know it is the problem of network. I came here [health facility] and reported this for them [health professionals]. (PWTB #11, Smart Pillbox)*

In other cases, a repeated or random alarm from the smart pillbox prompted multiple opening of the device.

*One day, after taking the medicine, it started alarming, so I opened it more than one time that day in order not to be disturbed. But it didn't stop and shout again and I opened it again. This happened only one day. (PWTB #8, Smart Pillbox Arm)*

Other PWTB also experienced a random alarm from the box before or after the scheduled time of taking their medication.

*For some time, it alarmed at 7:00 am which is before the adjusted time [the schedule was at 10:00am]. This was not right. If it was reminding me at 9:40 or 10 minutes later, it could be useful for me. But in this case, it is only disturbing. (PWTB #9, Smart Pillbox Arm)*

Due to network issues, PWTB in the label arm reported delays when receiving a confirmation message after they took their medications and submitted a text to the platform. As such, PWTB in the medication label arm were sometimes forced to send multiple messages when they were unable to receive immediate confirmation. This has created emotional discomfort and annoyance in some PWTB. However, during their next health facility appointment, PWTB were able to discuss the issue with HCWs who became aware of the problem.

*For I didn't receive a feedback message, I kept frequently sending of a message by assuming the message did not reach them [health professionals]. While I came here [in the health facility] on the 3rd month, I asked them why sometimes their message not sent back to me? (PWTB #22, Medication Label Arm)*

*There were issues I faced sometimes. I faced things like when I text, I wouldn't get a reply. Sister [name of focal nurse deleted] said, "it was a network problem". (PWTB #5, Medication Label Arm)*

There are PWTB who received multiple follow up text messages despite medication had been taken on time. When there were network issues, the platform is unlikely to receive the

notification that was sent when the smart pillbox was opened or when patients sent the "code" in the medication label arm. As such, PWTB often received reminder text messages notifying them that they had forgotten to take their medication. This led to frustration in PWTB because they feared the health professional may consider them to be non-adherent. Some even decided to come to the health facility before their appointments to explain the situation to the health professionals.

> *Sometimes I receive a message that says, "you did not take your medicine today". At that time, I came here [to the health facility] and explained that I have taken the medicine, but my house has a network problem. This inconvenience happens because of network, not because I didn't take it. (PWTB #11, Smart Pillbox)*

> *There are four or five messages in my phone saying "please take your medicine today" while I already have taken it. I don't know if they can fix these things with the box. But I did not miss it. (PWTB #1, Smart Pillbox Arm)*

## Theme 2: Lack of inclusivity (barrier)

As a barrier, HCWs and stakeholders identified that DAT lacks inclusion of all categories of PWTB. For example, those with no literacy, did not have phones, homeless and those in prison may not have the opportunity to use DATs.

> *When I was visiting [health facilities], patients who are using directly observed therapy (DOT) request to join DATs intervention. But I didn't face DAT user requesting to use DOT. To make it fair, people who request the technology must have an access (STK #1).*

> *There are patients who cannot text, but we talk to their relatives at home to show them how to text. Most patients get used to it after a few days. Those who are unable, they have to visit the health facility (HCW #17).*

> *Patients who don't have accommodation can't stay for a year, so I don't include them in the DAT program. However, they want to be part of the program. I always wonder how to handle such cases. (HCW #2).*

> *It is good to involve more partners, to mobilize more resource, to think more about scale up and inclusivity, to try it in other regions, and on different settings. It is important to try it in prison, to try it in pastoral communities, and try it in different settings (STK #3).*

## Theme 3: Fear of stigma due to unintended disclosure of TB status while using DATs (barrier)

Our study identified mixed perspectives about stigma related to the use of DATs. While stigma was not a concern for PWTB, HCWs reflected their fears (perceived stigma).

> *I am not afraid of what people might say about it [smart pillbox]. I am not afraid, usually I hold it in a plastic bag and sometimes I hold it as it is, but I do not mind what others think about it even if they know my TB status. It's not a concern for me. (PWTB #6, Smart Pillbox Arm)*

> *Mostly people perceive it [the smart pillbox] as a lunch box. But if they ask me about it, I tell them that it is my medicine box for my treatment follow-up. When I come here [to the health*

*facility] to receive the medicine, individuals see the smart pillbox and ask me "are you travelling with your lunch box?" But I tell them that it is a smart pillbox which helps me to follow-up the medication or treatment. (PWTB #9, Smart Pillbox Arm)*

According to HCWs' accounts, there were PWTB who were not comfortable when they were being home visited due to fear of stigma.

*Patients do not want home visits by healthcare workers because their families may not be unaware of their TB status (HCW #19)*

## Discussion

The overall finding suggests that DATs are highly acceptable among the diverse categories of participants in the presence of critical barriers that limit the implementation and use of DATs. In agreement with previous studies, our findings identified several reasons attributed to the high uptake of DATs by PWTB and healthcare providers [6,7,13,18,19]. Both PWTB and healthcare workers agreed that the use of DATs are relevant to improving adherence by reducing burdens related to frequent health facility visits, enhancing patient-provider relationship, increasing treatment flexibility, and reducing workload and work-related TB exposure.

Our finding, however, is in contrary to some studies which reported that DATs hampered patient-provider relationships more than conventional care [20,21]. These studies reported that DATs reduced the in-person interactions between PWTB and health professionals and contributed to less patient satisfaction and poor adherence in the context of an app-only intervention. While reduced in-person interactions were evident, our study highlighted that both PWTB and healthcare workers felt strongly connected by using DATs leading to more personalised care. The DATs helped health professionals closely monitor the treatment adherence of each PWTB, creating an opportunity to provide better patient-centered care through SMS, phone calls and home visits depending on individual patient adherence profiles.

While PWTB in this study reported they have no fear of stigma by using DATs, findings of previous studies were mixed concerning participants' perceptions surrounding stigma. For example, a study in India reported that issues of stigma related to the use of DATs contributed to lower adherence as PWTB were particularly concerned about travelling with the DAT during their clinic visits or when friends or relatives visited their home [17]. On the other hand, Rabinovich et al. identified stigma was not reported as a significant concern among PWTB in Cambodia [20]. The variations in these findings can be explained by the fact that stigma is a social determinant of health which is shaped and promulgated by institutional and community norms and interpersonal attitudes [22].

Unlike PWTB, HCWs in our study indicated concerns related to unintended disclosure of TB status that could lead to stigma which is largely in agreement with previous study findings [6,7,17]. As indicated by HCWs in this study, stigma may affect PWTB who did not disclose their TB status to family members or others. For instance, home visits by healthcare workers as part of the differentiated care can result in unintended disclosure of TB status and in return, this may lead to stigma. However, much detail is not available in our study about the actual experience of PWTB as none of them were home visited by healthcare workers. PWTB have shared their experiences of receiving only text messages and/or phone calls when they missed doses or dosing time.

While it was not commonly reported in previous studies, respondents in this study favourably identified that DATs enable PWTB to take their medications at a time and place of their

choosing. DATs also enabled healthcare workers to have more flexibility in the context of time and place to monitor the adherence profiles of their clients with TB and provide a differentiated response. This made DATs very attractive to both PWTB and healthcare workers and created a more person-centered treatment alternative than the usual directly observed therapy.

Our findings largely support prior studies demonstrating that poor network connectivity (mobile signals) was identified as a key barrier to the uptake of DATs [6,13,20]. But again, healthcare workers and stakeholders were positive about the implementation feasibility and adoption of DATs in the Ethiopian context given that there are opportunities including the rapid expansion of telecom services, phone usage, and electricity supply in Ethiopia. Furthermore, stakeholders also felt that the presence of TB care coordination platforms at national and regional levels, health digitization initiatives by the MOH, and the commitment of the MOH towards funding and supporting TB programs were important facilitating factors for the national adoption of DATs.

## Implications

The findings of the study suggest that DATs are highly acceptable among PWTB, HCWs, and policy stakeholders in Ethiopia. DATs were found to improve treatment flexibility, reduce the burden of frequent health facility visits, and enhance patient-provider relationships. However, key barriers to the implementation of DATs include poor infrastructure, lack of inclusivity for all PWTB, and concerns about stigma. Despite these challenges, several opportunities and initiatives, including the presence of TB care coordination platforms at national and regional levels, health digitization initiatives by the MOH, and the MOH's commitment to funding new TB programs, could support the feasible implementation, adaptation, and scale-up of DATs in the Ethiopian context. Future research should focus on developing strategies to overcome these barriers, particularly by addressing infrastructure challenges and ensuring the inclusivity of DATs for all PWTB, to maximize their effectiveness and sustainability in different settings.

## Strengths and limitations

A strength of our study includes the capturing of comprehensive perspectives of participants by engaging diverse groups of respondents including people with drug-sensitive TB, people with multi-drug resistant TB, healthcare workers and policy stakeholders in Ethiopia. Another strength of this study is that it adds more insights to the emerging body of knowledge regarding the role of acceptability in digital adherence technologies that can be used to help improve adherence to TB treatment in the future. However, findings in this study need to be interpreted in the context of some key limitations. As with all qualitative research, the findings cannot be considered representative of the broader views of other PWTB in Ethiopia. For instance, some groups of PWTB were excluded from the trial including homeless people and people with extrapulmonary TB. All health facilities were in urban areas, similar to the trial setting, so we cannot generalise to experiences of PWTB and HCWs from rural settings. Social desirability may have influenced PWTB and HCWs while reporting on the usefulness of DATs. In addition, only a small number of respondents were interviewed among key policy stakeholders which may limit our ability to highlight the national opportunities to integrate and scale-up DATs in the Ethiopian context. A further limitation of this study is the absence of input from relatives of persons with TB, who could have provided additional insights.

## Conclusions

In general, findings from this qualitative study have provided a rich set of perspectives relevant to policymakers in identifying the facilitators and barriers to the uptake of DATs in Ethiopia.

The overall finding suggests that DATs are highly acceptable among PWTB, HCWs and policy stakeholders. PWTB and HCWs identified several facilitating factors that increased the uptake of DATs including their usefulness and simplicity of use. On the other hand, critical barriers that limit the uptake of DATs were identified as poor infrastructure, lack of inclusiveness, and fear of stigma. Despite those barriers, key policy stakeholders believe there are several opportunities and initiatives that could support the feasible implementation, adaptation, and scale-up of DATs in the Ethiopian context, including the MOH's commitment to expanding health digitization efforts and funding new TB programs.

## Supporting information

**S1 Appendix. Characteristics of participants.**
(DOCX)

**S2 Appendix. Interview Guides.**
(DOCX)

## Acknowledgments

We would like to thank the participants who participated in this study, the National Tuberculosis Program at the Ministry of Health Ethiopia, the Disease Prevention and Control team at the regional health bureau of Addis Ababa City and Oromia regional state and woreda health offices, and the TB focal persons at each facility in implementing the ASCENT project.

## Author Contributions

**Conceptualization:** Zewdneh Shewamene, Degu Jerene, Katherine Fielding, Amare W. Tadesse.

**Data curation:** Mahilet Belachew, Amanuel Shiferaw, Mamush Sahlie, Demekech Gadissa, Tofik Abdurhman.

**Formal analysis:** Zewdneh Shewamene, Liza De Groot, Tanyaradzwa Dube, Amare W. Tadesse.

**Funding acquisition:** Degu Jerene, Katherine Fielding.

**Investigation:** Ahmed Bedru, Taye Leta, Katherine Fielding, Amare W. Tadesse.

**Methodology:** Zewdneh Shewamene, Natasha Deyanova, Katherine Fielding, Amare W. Tadesse.

**Project administration:** Mahilet Belachew, Amanuel Shiferaw, Mamush Sahlie, Demekech Gadissa, Tofik Abdurhman, Ahmed Bedru, Taye Leta, Amare W. Tadesse.

**Supervision:** Degu Jerene, Katherine Fielding, Amare W. Tadesse.

**Validation:** Liza De Groot, Tofik Abdurhman, Tanyaradzwa Dube, Natasha Deyanova, Degu Jerene, Katherine Fielding, Amare W. Tadesse.

**Writing – original draft:** Zewdneh Shewamene.

**Writing – review & editing:** Katherine Fielding, Amare W. Tadesse.

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
