## [Decision Letter · Decision Letter 0]

28 May 2024

PDIG-D-23-00479

Facilitators and barriers to uptake of digital adherence technologies in improving TB treatment adherence in Ethiopia: A qualitative study

PLOS Digital Health

Dear Dr. Shewamene,

Thank you for submitting your manuscript to PLOS Digital Health. After careful consideration, we feel that it has merit but does not fully meet PLOS Digital Health's publication criteria as it currently stands. Therefore, we invite you to submit a revised version of the manuscript that addresses the points raised during the review process.

Please submit your revised manuscript within 30 days Jun 27 2024 11:59PM. If you will need more time than this to complete your revisions, please reply to this message or contact the journal office at digitalhealth@plos.org. Please include the following items when submitting your revised manuscript:

We look forward to receiving your revised manuscript.

Kind regards,

Margaret Isioma Ojeahere, MBBS, FWACP

Academic Editor

PLOS Digital Health

Journal Requirements:

1. Please amend your detailed online Financial Disclosure statement. This is published with the article. It must therefore be completed in full sentences and contain the exact wording you wish to be published.

a) State the initials, alongside each funding source, of each author to receive each grant. For example: "This work was supported by the National Institutes of Health (####### to AM; ###### to CJ) and the National Science Foundation (###### to AM)."

2. Please provide separate figure files in .tif or .eps format only and remove any figures embedded in your manuscript file. Please also ensure that all files are under our size limit of 10MB. You may leave the figure captions or legends in the manuscript.

For more information about how to convert your figure files please see our guidelines: https://journals.plos.org/digitalhealth/s/figures

Additional Editor Comments (if provided):

Reviewer One: 

The content of the paper is quite interesting, novel and relevant for TB treatment & digital adherence technologies. The main claims of the paper were made clear and a great amount of evidence was included in the paper. Unfortunately, the delivery of the collected data is lacking formality and correct grammar in multiple instances (changes pointed out specifically down below). Along with that the formatting and font/font sizes, headings etc. differ throughout. Besides, the results section would greatly benefit from a more precise display, it is fairly easy to get lost throughout all the different points presented. The discussion would also benefit from implementing future recommendations for practice and research. Finally, it would be interesting to display study protocols, interview guidelines and additional material used during the research period under supporting information. By applying the changes listed down below, this qualitative paper will make for a great paper! 

• Title Page, p.1. The font of affiliation #5 does not match with the other font used

• Author summary, p. 3 Font seems to be in grey 

• Introduction, p.4, 66-67, wording/phrasing sounds informal, I would take out (excluding COVID-19) as emphasis is put on “first leading infectious killer”, remains true regardless of the appearance of COVID-19 

• Study Design & Participants, p.5, 103 Please specifiy who is meant with key stakeholders (e.g. religious/community/ etc. leaders?)

• Study Design & Participants, p.5, 112, Incorrect spelling: take out “A” at the beginning of the sentence 

• Interview process, p. 6, 116, incorrect spelling: degrees -> plural 

• Interview process, p.6, 116: Has there been adequate training/ assistance concerning the cultural background of the study setting?

• Data Analysis, p.6, Font size differs?

• Data Analysis, p.6-7, elaborate on pseudonymization or anonymization? Elaborate on whether transcripts etc. were securely stored, when transcripts and notes will be deleted etc. 

• Ethical considerations, p.7, 153, informal spelling: can’t -> cannot

• Characteristics of participants, p.8, 165, incorrect phrasing: include: The majority OF healthcare workers 

• Facilitators of DATs uptake, p.11, 199, What does XX stand for at the end of the sentence? 

• Facilitators of DATs uptake, p.13 Formatting is off

• Facilitators of DATs uptake, p. 15, 276, font and font size differs

• Reduce workload and work-related TB exposure to providers, p.16, 294: missing letter: ReduceD, 295, missing word, add “a” few

• Theme 2, p.16, 306, Heading is misspelled, is it supposed to mean Easy for use? 

• Theme 3, p.18, 351, Wording: important others? Are they referring to “important significant others”?

• Theme 3, p.18, 355, wording, change to: “in their TB treatment”

• Theme 4, p. 19, 362, missing word: can increase “THE” uptake

• Overall heading stile is not coherent 

• Barriers for uptake of DATS, p.20, 390, missing word: those “WHO” did not have phones, 392, missing word: to “THE” implementation, 394, missing word: can result “IN THE” fear of stigma 

• Theme “1” Network and electricity issues, p.21, 397, Duplicate Heading, Theme one already exists, maybe it is best to change it to Theme 1.2. ? 404, wrong spelling: my? Do they mean: may?, 428, PWTB is always addressed singularly, needs to be plural, add an “s”, 447, missing word: they “HAD” forgotten.., 448: missing words, may consider them “TO BE” non-adherent, come to “THE” health facility

• Theme 2: Lack of inclusivity, p. 24, 459,missing word: As “A” barrier, 460 odd sentence: those did not have phones?, people in prison: is that not too far-fetched since this target group is not included here?

• Theme 3: Fear of Stigma etc., p.24, 473, missing word: was 

• Discussion, p.25, 495-496, phrasing is odd

• Strenghts and limitations, p. 27, 537, missing word “THE” capturing, 538, missing letter: engaging diverse “groupS”, including “DRUG SENSITIVE PEOPLE”, 50 is “THAT” it adds, 547: DATS font is different

Reviewer Two

General comment

The manuscript has important piece of information and crucial to the field that gives a hope on the impact of digital technology in helping the monitoring of medication adherence to TB treatment in Ethiopia, however, access to technology and utilization of digital platform is still limited in rural part of the country where TB is highly prevalent.

Specific comments

Suggested Title: good to minimize double use of key terms, remove ‘adherence’: “Facilitators and barriers to uptake of digital technologies in improving TB treatment adherence in Ethiopia: A qualitative study”

Abstract

- Page 2: line 36-38: ‘Network issue’ is appeared as barrier, but it is not specific and may not be clear for readers. Do you mean network is not clear, patient network or internet connectivity?

Background

- Well described except, mix of acronym and long form inconsistently eg. DOTs and DATs

- Good to add some literature that shows the magnitude of TB and treatment adherence level in Ethiopia

Methods

- As this study is nested from cluster RCT study, did this qualitative study consider both the intervention and comparison group perspectives about the technology use to track TB treatment?

- The reason for the purposively focusing ‘Urban setting’ is not clear at all. Although TB is common every part of the country, but disproportionately higher in Rural areas and adherence to treatment is also affected by several barriers in Rural setting. So why this study only targets Urban dwellers. Authors should give tips about this issue.

- Categories of study participants: Healthcare workers are also key stakeholders. Kindly check this one as well.

- Page 6: line 114-121: How was the interview conducted? Where was the interview held, at Health facility or home?

Results: There should be a section to separate the methods and results

- Good to move the description of ‘Themes’ to Methods section

- Improved patients-provider relationship through person-centred care… suggestion “Improved patients-provider relationship”

- Page 16: Split the sub-topic: as ‘Reduce Healthcare providers workload’ ‘Reduce Health care providers work-related TB exposure’

- Good to have brief summary about the ‘implication of the study’ that might be good for high TB burden countries and being challenged by non- adherence issue.

Reviewer Three:

I would like to commend the authors on carrying out this study which I believe will contribute to the body of knowledge in Tuberculosis treatment. The study also highlights innovative approaches to ensuring adherence in patients.

I however have the following recommendations.

Introduction: Authors should ensure they discuss to some extent what the DOT is. There appears to be no attempt at educating the reader about DOT.

Study setting: The study focused more on urban settings. No mention of a rural setting was made. Is Oromia rural? Was the mention of Oromia in this sentence as an urban area an error?

“The study sites in Addis Ababa were all urban while predominantly urban in Oromia”

The authors should please make an attempt to describe the Everwell platform in a bid to educate the reader.

Study design and participants: An additional group composed of relatives of persons with TB would have served as an important demographic to consider for the Interview process.

As the training for the research assistants lasted for a day, could you please share the degree of agreement between the interviewers and the trainees?

Did you have mock trials on more participants to see if the research assistants acquired the skills for data collection?

How did you ensure inter-rater variation?

In addition, a pilot study may also have helped to guarantee that the research assistants after the training were capable of conducting the study.

Proper description of 'smart pill box' and the 'medication labels' could have been described in the methodology.

I will suggest that the process of monitoring patients' use of medication be described in detail in the methodology. 

Please share what objective method was used to confirm that the patients’ actually used their medication beyond the technological methods of sending a text message?

Thank you.

Reviewers' comments:

Reviewer's Responses to Questions

**Comments to the Author**

1. Does this manuscript meet PLOS Digital Health’s publication criteria? Is the manuscript technically sound, and do the data support the conclusions? The manuscript must describe methodologically and ethically rigorous research with conclusions that are appropriately drawn based on the data presented.

Reviewer #1: Yes

Reviewer #2: Yes

Reviewer #3: Yes

2. Has the statistical analysis been performed appropriately and rigorously?

Reviewer #1: N/A

Reviewer #2: Yes

Reviewer #3: Yes

3. Have the authors made all data underlying the findings in their manuscript fully available (please refer to the Data Availability Statement at the start of the manuscript PDF file)?

Reviewer #1: No

Reviewer #2: No

Reviewer #3: Yes

4. Is the manuscript presented in an intelligible fashion and written in standard English?

Reviewer #1: No

Reviewer #2: Yes

Reviewer #3: Yes

5. Review Comments to the Author

Reviewer #1: The content of the paper is quite interesting, novel and relevant for TB treatment & digital adherence technologies. The main claims of the paper were made clear and a great amount of evidence was included in the paper. Unfortunately, the delivery of the collected data is lacking formality and correct grammar in multiple instances (changes pointed out specifically down below). Along with that the formatting and font/font sizes, headings etc. differ throughout. Besides, the results section would greatly benefit from a more precise display, it is fairly easy to get lost throughout all the different points presented. The discussion would also benefit from implementing future recommendations for practice and research. Finally, it would be interesting to display study protocols, interview guidelines and additional material used during the research period under supporting information. By applying the changes listed down below, this qualitative paper will make for a great paper! Specific comments can be found in the attached file

Reviewer #2: Reviewer report 

To: PLOS Digital Health Journal 

Title: Facilitators and barriers to uptake of digital adherence technologies in improving TB treatment adherence in Ethiopia: A qualitative study 

General comment 

The manuscript has important piece of information and crucial to the field that gives a hope on the impact of digital technology in helping the monitoring of medication adherence to TB treatment in Ethiopia, however, access to technology and utilization of digital platform is still limited in rural part of the country where TB is highly prevalent. 

Specific comments 

Suggested Title: good to minimize double use of key terms, remove ‘adherence’: “Facilitators and barriers to uptake of digital technologies in improving TB treatment adherence in Ethiopia: A qualitative study”

Abstract

- Page 2: line 36-38: ‘Network issue’ is appeared as barrier, but it is not specific and may not be clear for readers. Do you mean network is not clear, patient network or internet connectivity? 

Background

- Well described except, mix of acronym and long form inconsistently eg. DOTs and DATs

- Good to add some literature that shows the magnitude of TB and treatment adherence level in Ethiopia

Methods

- As this study is nested from cluster RCT study, did this qualitative study consider both the intervention and comparison group perspectives about the technology use to track TB treatment? 

- The reason for the purposively focusing ‘Urban setting’ is not clear at all. Although TB is common every part of the country, but disproportionately higher in Rural areas and adherence to treatment is also affected by several barriers in Rural setting. So why this study only targets Urban dwellers. Authors should give tips about this issue.

- Categories of study participants: Healthcare workers are also key stakeholders. Kindly check this one as well.

- Page 6: line 114-121: How was the interview conducted? Where was the interview held, at Health facility or home? 

Results: There should be a section to separate the methods and results

- Good to move the description of ‘Themes’ to Methods section

- Improved patients-provider relationship through person-centred care… suggestion “Improved patients-provider relationship”

- Page 16: Split the sub-topic: as ‘Reduce Healthcare providers workload’ ‘Reduce Health care providers work-related TB exposure’

- Good to have brief summary about the ‘implication of the study’ that might be good for high TB burden countries and being challenged by non- adherence issue.

Reviewer #3: ARTICLE REVIEW- 

Facilitators and barriers to uptake of digital adherence technologies in improving TB treatment adherence in Ethiopia: A qualitative study

I would like to commend the authors on carrying out this study which I believe will contribute to the body of knowledge in Tuberculosis treatment. The study also highlights innovative approaches to ensuring adherence in patients.

I however have the following recommendations.

Introduction: Authors should ensure they discuss to some extent what the DOT is. There appears to be no attempt at educating the reader about DOT.

Study setting: The study focused more on urban settings. No mention of a rural setting was made. Is Oromia rural? Was the mention of Oromia in this sentence as an urban area an error?

“The study sites in Addis Ababa were all urban while predominantly urban in Oromia”

The authors should please make an attempt to describe the Everwell platform in a bid to educate the reader.

Study design and participants:  An additional group composed of relatives of persons with TB would have served as an important demographic to consider for the Interview process.

As the training for the research assistants lasted for a day, could you please share the degree of agreement between the interviewers and the trainees?

Did you have mock trials on more participants to see if the research assistants acquired the skills for data collection?

How did you ensure inter-rater variation?

In addition, a pilot study may also have helped to guarantee that the research assistants after the training were capable of conducting the study.

Proper description of 'smart pill box' and the 'medication labels' could have been described in the methodology.

I will advise that the process of monitoring patients' use of medication be described in detail in the methodology. 

Please share what objective method was used to confirm that the patients’ actually used their medication beyond the technological methods of sending a text message?

Thank you.

6. PLOS authors have the option to publish the peer review history of their article (what does this mean?). If published, this will include your full peer review and any attached files.

**Do you want your identity to be public for this peer review?** For information about this choice, including consent withdrawal, please see our Privacy Policy.

Reviewer #1: Yes: Sophie Stützle

Reviewer #2: No

Reviewer #3: No

---

## [Decision Letter · Decision Letter 1]

23 Sep 2024

PDIG-D-23-00479R1

Facilitators and barriers to uptake of digital adherence technologies in improving TB care in Ethiopia: a qualitative study

PLOS Digital Health

Dear Dr. Shewamene,

Thank you for submitting your manuscript to PLOS Digital Health. After careful consideration, we feel that it has merit but does not fully meet PLOS Digital Health's publication criteria as it currently stands. Therefore, we invite you to submit a revised version of the manuscript that addresses the points raised during the review process.

Please submit your revised manuscript within 30 days Oct 23 2024 11:59PM. If you will need more time than this to complete your revisions, please reply to this message or contact the journal office at digitalhealth@plos.org. Please include the following items when submitting your revised manuscript:

We look forward to receiving your revised manuscript.

Kind regards,

Margaret Isioma Ojeahere, MBBS, FWACP

Academic Editor

PLOS Digital Health

Journal Requirements:

Additional Editor Comments (if provided):

Reviewers' comments:

Reviewer's Responses to Questions

**Comments to the Author**

1. If the authors have adequately addressed your comments raised in a previous round of review and you feel that this manuscript is now acceptable for publication, you may indicate that here to bypass the “Comments to the Author” section, enter your conflict of interest statement in the “Confidential to Editor” section, and submit your "Accept" recommendation.

Reviewer #1: All comments have been addressed

Reviewer #3: All comments have been addressed

Reviewer #4: All comments have been addressed

2. Does this manuscript meet PLOS Digital Health’s publication criteria? Is the manuscript technically sound, and do the data support the conclusions? The manuscript must describe methodologically and ethically rigorous research with conclusions that are appropriately drawn based on the data presented.

Reviewer #1: Partly

Reviewer #3: Yes

Reviewer #4: Yes

3. Has the statistical analysis been performed appropriately and rigorously?

Reviewer #1: N/A

Reviewer #3: Yes

Reviewer #4: N/A

4. Have the authors made all data underlying the findings in their manuscript fully available (please refer to the Data Availability Statement at the start of the manuscript PDF file)?

Reviewer #1: Yes

Reviewer #3: Yes

Reviewer #4: Yes

5. Is the manuscript presented in an intelligible fashion and written in standard English?

Reviewer #1: Yes

Reviewer #3: Yes

Reviewer #4: Yes

6. Review Comments to the Author

Reviewer #1: The research article has been properly adapted to the reviewers comments. I have added a few minor changes as comments in the attached pdf revision version. I would ask to please check the general submission guidelines provided by the journal (an example can be found on the official website). This concerns the font size, the headings for the figures etc.

Reviewer #3: A significant amount of the recommendations have been attended to.

Reviewer #4: 1. page 5, line 90-92. sustainability should be changed to "sustainably"

2. page 30, line 600-601. Punctuation to be done properly by inserting comma. "Oromia regional, state....).

 Woreda health officer changed to "ward health officers"

7. PLOS authors have the option to publish the peer review history of their article (what does this mean?). If published, this will include your full peer review and any attached files.

**Do you want your identity to be public for this peer review?** For information about this choice, including consent withdrawal, please see our Privacy Policy. 

Reviewer #1: Yes: Sophie Stützle

Reviewer #3: Yes: 

Reviewer #4: Yes: NWOGA CHARLES NNAEMEKA

---

## [Decision Letter · Decision Letter 2]

13 Oct 2024

Facilitators and barriers to uptake of digital adherence technologies in improving TB care in Ethiopia: a qualitative study

PDIG-D-23-00479R2

Dear Dr. Shewamene,

We are pleased to inform you that your manuscript 'Facilitators and barriers to uptake of digital adherence technologies in improving TB care in Ethiopia: a qualitative study' has been provisionally accepted for publication in PLOS Digital Health.

Best regards,

Margaret Isioma Ojeahere, MBBS, FWACP

Academic Editor

PLOS Digital Health

Reviewer Comments (if any, and for reference):

Reviewer's Responses to Questions

**Comments to the Author**

1. If the authors have adequately addressed your comments raised in a previous round of review and you feel that this manuscript is now acceptable for publication, you may indicate that here to bypass the “Comments to the Author” section, enter your conflict of interest statement in the “Confidential to Editor” section, and submit your "Accept" recommendation.

Reviewer #1: All comments have been addressed

Reviewer #4: All comments have been addressed

2. Does this manuscript meet PLOS Digital Health’s publication criteria? Is the manuscript technically sound, and do the data support the conclusions? The manuscript must describe methodologically and ethically rigorous research with conclusions that are appropriately drawn based on the data presented.

Reviewer #1: Yes

Reviewer #4: Yes

3. Has the statistical analysis been performed appropriately and rigorously?

Reviewer #1: Yes

Reviewer #4: N/A

4. Have the authors made all data underlying the findings in their manuscript fully available (please refer to the Data Availability Statement at the start of the manuscript PDF file)?

Reviewer #1: Yes

Reviewer #4: Yes

5. Is the manuscript presented in an intelligible fashion and written in standard English?

Reviewer #1: Yes

Reviewer #4: Yes

6. Review Comments to the Author

Reviewer #1: All comments have been addressed properly. Nonetheless I would ask the authors to check the submission guidelines of the journal again in terms of formatting.

Reviewer #4: (No Response)

7. PLOS authors have the option to publish the peer review history of their article (what does this mean?). If published, this will include your full peer review and any attached files.

**Do you want your identity to be public for this peer review?** For information about this choice, including consent withdrawal, please see our Privacy Policy.

Reviewer #1: No

Reviewer #4: **Yes: **NWOGA CHARLES NNAEMEKA
